# Theoretical Study of Zirconium Isomorphous Substitution into Zeolite Frameworks

**DOI:** 10.3390/molecules24244466

**Published:** 2019-12-05

**Authors:** Duichun Li, Bin Xing, Baojun Wang, Ruifeng Li

**Affiliations:** 1College of Chemistry and Chemical Engineering, Taiyuan University of Technology, Taiyuan 030024, China; liduichun1985@163.com (D.L.); xingbin@tyut.edu.cn (B.X.); 2Key Laboratory of Coal Science and Technology of Ministry of Education and Shanxi Province, Taiyuan University of Technology, Taiyuan 030024, China

**Keywords:** density functional theory, zeolite, isomorphous substitution, distribution of Zr, structural stability, Brönsted acidity

## Abstract

Systematic periodic density functional theory computations including dispersion correction (DFT-D) were carried out to determine the preferred location site of Zr atoms in sodalite (SOD) and CHA-type topology frameworks, including alumino-phosphate-34 (AlPO-34) and silico-alumino-phosphate-34 (SAPO-34), and to determine the relative stability and Brönsted acidity of Zr-substituted forms of SOD, AlPO-34, and SAPO-34. Mono and multiple Zr atom substitutions were considered. The Zr substitution causes obvious structural distortion because of the larger atomic radius of Zr than that of Si, however, Zr-substituted forms of zeolites are found to be more stable than pristine zeolites. Our results demonstrate that in the most stable configurations, the preferred favorable substitutions of Zr in substituted SOD have Zr located at the neighboring sites of the Al-substituted site. However, in the AlPO-34 and SAPO-34 frameworks, the Zr atoms are more easily distributed in a dispersed form, rather than being centralized. Brönsted acidity of substituted zeolites strongly depends on Zr content. For SOD, substitution of Zr atoms reduces Brönsted acidity. However, for Zr-substituted forms of AlPO-34 and SAPO-34, Brönsted acidity of the Zr-O(H)-Al acid sites are, at first, reduced and, then, the presence of Zr atoms substantially increased Brönsted acidity of the Zr-O(H)-Al acid site. The results in the SAPO-34-Zr indicate that more Zr atoms substantially increase Brönsted acidity of the Si-O(H)-Al acid site. It is suggested that substituted heteroatoms play an important role in regulating and controlling structural stability and Brönsted acidity of zeolites.

## 1. Introduction

Zeolites are crystalline microporous aluminosilicates with periodically arranged cages and channels that have wide industrial applications as catalysts, adsorbents, and ion exchangers. To improve particular catalytic performance, a particular effective method is to modify a zeolite to adapt requirements of different catalytic reactions, and this involves modifying zeolite frameworks via isomorphous substitution of their tetrahedral sites using transition metal cations [1]. These metal ions modify the structure, acidity, and catalytic behavior of zeolites and play a significant role in designing catalytic and adsorptive properties of zeolites.

Isomorphous substitution of tetrahedral sites using transition metal ions is an interesting and important idea for the design of novel catalysts. Researchers have been working on the design of new types of zeolite frameworks with unique pore topologies and functionalities for specific applications [2]. Elements of different groups (III, IV, V, VI, and VII) have been introduced into zeolite frameworks. It has been shown that the local coordination environment around the positions where metal ions are incorporated have a great effect on the structure and acidity of the zeolites, and therefore on the catalytic reactivity [3,4,5,6,7,8,9]. Therefore, it is important to investigate the relationships among location, stability, and structural properties of different heteroatoms in frameworks [1]. For example, isomorphous substitution with trivalent elements (such as B, Ga, and Fe) has been used instead of common Al^3+^ to modify corresponding structural properties and acidity [10]. MAPO-34 (M = Fe, Co, Ni, La, Ce) was synthesized and presented the same CHA crystalline structures as those of the SAPO-34 framework. NH_3_ temperature programmed desorption (NH_3_-TPD) and Fourier transform infrared spectrometer (FTIR) spectral results have indicated that incorporating metal ions causes weaker acidity [11], and also, incorporating a small amount of a transition metal into the SAPO-34 structure causes significant changes to the acid properties [12]. In general, introducing these cations into a framework is not always easy and reasonable, and only a few transition metal-substituted zeolites have been characterized [13,14,15,16,17,18,19,20,21]. 

Substituting aluminum for a tetrahedral silicon in the aluminosilicate zeolite and substituting silicon for a tetrahedral phosphorous in the aluminophosphate zeolite introduce a negative charge in the framework; charge compensation with a proton creates a catalytically active Brönsted acid site. The bridging hydroxyl group is the source of the Brönsted acid site. Acid properties (such as number, strength, location, and distribution) can be adjusted. Different amounts of Si and Al atoms in the framework result in different acid sites and properties. Many works have focused on modifications of zeolite catalysts, which change the reactivity of acid-active site through adjusting the acid properties. Many studies have reported the properties of modified zeolite catalysts via the incorporation of Zr atoms into frameworks [1,22,23,24,25,26,27,28,29,30,31,32]. Experimentally, it is difficult to characterize the specific location and distribution of Zr atoms in a framework, as well as the stability and acidity of the corresponding structures. Hence, clarifying the influences of metal incorporation and corresponding changes in structural stability and acid properties is the motivation for this study. Theoretical calculations are a good tool to employ for systematically studying these questions. Using tetrahedral coordinates that have already been determined by various experiments, a reasonable model can be built to simulate the zeolite framework. Because the location and distribution of heteroatoms in the framework are connected to the structural properties of zeolites, theoretical studies give a better understanding of the nature of isomorphically substituted zeolites [33,34].

In this work, we present a systematic periodic density functional theory study on the location of Zr heteroatoms in sodalite (SOD) and CHA-type topological including alumino-phosphate-34 and silico-alumino-phosphate-34 (AlPO-34/SAPO-34) and on the relative stability and Brönsted acidity of the corresponding structures. First, we attempt to clarify whether Zr-substituted forms of SOD and AlPO-34/SAPO-34 frameworks (termed SOD-Zr_n_, AlPO-34-Zr_n_, and SAPO-34-Zr_n_, in which “n” represents the number of Zr substitutes in a framework) are stable. Then, we investigate the influences of different amounts of Zr atoms on the stability of all of the substituted frameworks. In addition, we check the preferred location sites of Zr atoms in SOD and CHA-type topological frameworks. The substitution energy and local structural perturbation parameter (Ω) are used to analyze the structural stability and structural distortion in the SOD-Zr_n_, AlPO-34-Zr_n_, and SAPO-34-Zr_n_ frameworks. Deprotonation energy (DPE) is then used to characterize the corresponding Brönsted acidity of the most stable configurations with different amounts of Zr atoms in the SOD and AlPO-34/SAPO-34 frameworks. Our aims are to find a general rule for stable structures and to provide an atomistic picture of lattice distortion after Zr atoms are introduced into the SOD and AlPO-34/SAPO-34 frameworks. The computational results provide insight into the relationship between the location sites of Zr atoms in the framework and structural stability, as well as the relationship between the location sites of Zr atoms in the framework and the Brönsted acidity of substituted structures. These relationships should be useful as guidance in the design and synthesis of new zeolite frameworks.

## 2. Results and Discussion

### 2.1. Structural Stability of Isomorphous Substituted SOD

#### 2.1.1. Distribution of Al Atoms in SOD

To study the structural stability of Al-substituted SOD, the Al distribution in a SOD cage with a Si/Al ratio range from 11 to one is studied in detail. Different Si/Al ratios in SOD depend on the number of Al atoms that are substituted for Si atoms. According to the periodic SOD model, the number of Al substitutes is set from one to six to represent six different models with Si/Al ratios set as 11, five, three, two, 1.4, and one, respectively. All of the possible distributions of Al atom are calculated using Löwenstein’s rules [35]. A single Si atom in a unit cell of SOD was isomorphically substituted with one Al atom, and to avoid the influence of negative charge that is incorporated by Al substitution, the charge is compensated by a H proton during the calculation to provide a Brönsted acid. There are four O atoms around the Al-substituted site, and thus there are four possible locations for H (named H1, H2, H3, and H4) in the Al-substituted SOD. The distribution of H protons among these four sites is calculated. The results suggest that the relative energy differences for these four sites of H protons are less than 0.02 eV, and this is in good agreement with the observation reported by Neurock [36]. Neurock et al. found that there were no obvious differences among these four kinds of H protons because the relative energy differences of the four different location sites of the H protons were about 0.21–0.31 eV. Therefore, the distribution of H protons among these four kinds of sites is not considered during the calculations of Al distributions.

Structural stability strongly depends on the location of isomorphous substitutions. Relative energy differences (ΔE) are used to investigate the relative stability between structures with Al atoms at different sites to determine the most stable sites for Al substitution sites in the framework. The values of ΔE are calculated as differences from the total energy of the structure with respect to that of the different substitution sites of the Al atom. Then, the distribution of Al atoms in the SOD framework is obtained, and the most stable configurations for different Si/Al ratios are presented in Figure 1. Because the 12 T atoms in the SOD cages are equal, there is only one distribution for one and for six Al atoms in the SOD framework. As shown in Figure 1, all of the six Al atoms lie in diagonal positions of the four-membered ring in the framework for Si/Al = 1. However, according to Löwenstein’s rules, there are 14 possible combinations of structures with different Tn sites for the structures that have two Al atoms (Si/Al = 5). Plots of ΔE are shown in Figure 2; for these plots, the benchmark is the energy of the structure that has the combination T5 + T11 sites. As shown in Figure 2, the structure that has Al atoms at T5 + T11 is the preferred location for two Al atoms, whereas the structure with Al atoms at T6 + T12 is the most unfavorable location for two Al atoms. As shown from the distribution of the most stable combination of Tn sites shown in Figure 1, two Al atoms lie in the diagonal position of the same six-membered ring of the SOD cage. The distribution of Al atoms in SOD with Si/Al equals three and that with Si/Al equals two are determined. In addition, the corresponding values of ΔE for Al-substituted SOD with different Si/Al ratios is plotted (Figure 2). The most suitable positions for the distribution of three Al atoms are T2 + T5 + T9, and the most unsuitable positions are T6 + T11 + T12. For the distribution of four Al atoms in SOD, T2 + T5 + T6 + T9 are the most energetically favorable for substitution, and T5 + T6 + T11 + T12 are the least energetically favorable for substitution of four Al atoms. The most stable locations for Al atoms in the SOD framework with Si/Al equals three and Si/Al equals two are shown in Figure 1. As shown in Figure 1, three Al atoms lie in the meta position of the same six-membered ring of the SOD cage. For the locations of four Al atoms in the SOD framework, the sites of three of the four Al atoms are the same as that of the Al atoms in the structure with Si/Al = 3, and the fourth Al atom lies in the diagonal position of the four-membered ring that is adjacent to the six-membered ring in which the three Al atoms are located. In addition, there are only two different configurations for Al atoms in the SOD with Si/Al = 1.4, and the corresponding calculated results of the most stable configuration are listed in Table 1. As shown in Figure 1, the most stable location sites for five Al atoms in the SOD framework are the same as the distribution of Al atoms in the structure for which Si/Al equals one.

Therefore, these computational results indicate that the location of the heteroatom in the SOD framework is not random. From the distribution of Al atoms, we conclude that the Al atoms that are located in the diagonal position of the secondary building unit (four-membered ring and six-membered ring) are the most energetically favorable. When the number of Al atoms increases, they are likely to distribute in the diagonal positions of the four-membered ring in the SOD cage. For each Si/Al ratio, the structure that has the lowest energy is chosen for further investigations. The substitution energy (E_sub_) that corresponds to the energy change value of the substitution reaction is an appropriate criterion for evaluating the energetic priority for a specific T site substitution by a heteroatom. A smaller E_sub_ means that substitution occurs more easily. To compare the difficulty of substitutions with different content of Al or Zr atoms in the framework, the average substitution energy (E_av-sub_) is used in this work. The calculation equations are given in the Appendix A in detail. The substitution difficulty of Al atoms is then investigated, and the corresponding E_av-sub_ of the most stable configurations of the SOD framework with different Si/Al ratios are also listed in Table 1. As shown in Table 1, the value of E_av-sub_ decreases with a decrease in the Si/Al ratio, and this means that the SOD framework that has the lower Si/Al ratio is more easily synthesized because of the smaller value of E_av-sub_ and because of enhanced structural stability.

#### 2.1.2. Structural Stability of Al-Substituted SOD

Generally, the collapse of a zeolite begins at the Al sites in the framework [37]. Therefore, in the present work, the electrostatic potential (*ν*) of the Al-O bond in the SOD frameworks that have Si/Al ratios from 11 to one are investigated. The value of *ν* can be considered as a measure of the stability of the Al-O bond, that is, the stability of the structural framework. The calculation equations are given in the Appendix A in detail. The calculated values of *ν* are listed in Table 2, where it can be seen that there are no obvious differences in the values of *ν* for different Si/Al ratios. The calculated results suggest that the Al content in the framework has a negligible effect on structural stability.

### 2.2. Distribution of Zr and Structural Stability of Zr-Substituted SOD

To evaluate the preferred locations for Zr substitution of Si atoms in the SOD framework, the substitution energy of the Si atom substituted by Zr atom is investigated. The Si/Zr substitution energy is 1.84 eV, and this means that incorporating the first Zr atom into the SOD framework is more difficult than that of the first Al atom (0.17 eV). Therefore, the distribution of Zr atoms in the SOD framework is investigated on the basis of the most stable distribution of Al atoms in the SOD framework for Si/Al ratios from 11 to one, however, the distribution of Zr atoms in the pure siliceous SOD framework is not considered in this work.

#### 2.2.1. Stable Location of Zr atoms in SOD Framework of Si/Al Equals Eleven

On the basis of the SOD framework in which the Si/Al ratio is 11, the distributions of Zr atoms in the SOD are discussed. There are 11 possible T sites for the location of the first Zr atom in the SOD framework with Si/Al = 11, and these are named SOD-Zr_1_. To evaluate the most favorable tetrahedral site for Zr atom substitution in the SOD framework, the values ∆E for these 11 possible locations are investigated. The relative ∆E values for all of these 11 possible SOD-Zr_1_ structures are plotted in Appendix A in the Appendix A. The relative ∆E values are calculated from the total energies of the structures with respect to that of the most stable structure for the different T_n_ sites. The calculated Si/Zr substitution energy for the most stable location of Zr atoms in the SOD framework with Si/Al = 11 is −0.56 eV. The calculated results suggest that it is easier to substitute Zr atoms in the SOD framework with Si/Al = 11 as compared with the pure siliceous SOD framework. Appendix A in the Appendix A shows the most stable configuration of SOD-Zr_1_. As shown in Appendix A, the Zr and Al atoms are located in the meta positions of the same six-membered ring.

On the basis of the most stable configuration of SOD-Zr_1_, the distribution of the second Zr atom substitution is investigated. There are ten possible T sites for the location of the second Zr atom. The ∆E values of these ten possible locations are calculated and plotted, as shown in Appendix A, and the most stable configuration of SOD-Zr_2_ is shown in Appendix A, which shows that the most stable site for the second Zr atom is the neighboring site of the Al atom. Furthermore, two Zr atoms and one Al atom lie in the same six-membered ring, and two Zr atoms are distributed in the form of a Zr-O-Zr connection. In addition, the distributions of the third ~ sixth Zr atoms are studied and determined using the same rules introduced above. The ∆E values of the corresponding Zr-substituted SOD are plotted, as shown in Appendix A, and the most stable configurations of SOD-Zr_n_ (n = 3~6) are shown in Appendix A. The most favorable sites for the third ~ sixth Zr atoms are located around the Al atom, and all of these six Zr atoms lie in different six-membered rings adjacent to the four-membered ring in which the Al atom is located. Therefore, the distribution of Zr in the SOD framework forms a Zr-island structure. The distributions of Zr for structures with Si/Al ratios from five to one are determined using the same rules introduced above. According to ∆E values plotted in Appendix A, the most stable configurations for multiple Zr atoms substituted in the SOD framework are shown in Appendix A in the Appendix A. The most suitable positions for Zr atoms distributions in the SOD framework with Si/Al = 5, Si/Al = 3, Si/Al = 2, Si/Al = 1.4, and Si/Al = 1 are those in which the Zr atoms are located around Al atoms, however, the distribution of further substitutions of Zr atoms is not discussed. This attributed to the ionic radii of the Zr atom being larger than that of the Si atom, and this can directly affect the [SiO_4_] tetrahedra in terms of distortion with respect to the normal silica structure as a result of strains generated by the [ZrO_4_] tetrahedra in the framework.

The substitution difficulty of incorporating one and multiple Zr atoms into the SOD framework with Si/Al ratios from 11 to one is studied from the results obtained for the Si/Zr substitution energies of multiple Zr atoms. The calculated Si/Zr substitution energies are listed in Table 3; for these energies, the benchmark is the energy of the Al-substituted SOD framework that has Si/Al ratios from 11 to one. As listed in Table 3, the Zr-substituted SOD framework requires negative energy, and this reflects the structural stability of this framework. In terms of the influence of the Si/Al ratio on the substitution difficulty, the results indicate that when the Si/Al ratio is lower, incorporation of Zr into the SOD framework is easier. With respect to the substitution energies for which the benchmark is the energy of the Al-substituted SOD, it can be concluded that formation of Zr-substituted SOD frameworks is thermodynamically favorable, and this explains the difficulty of synthesizing Zr-containing SOD. Furthermore, the most favorable sites for substituting Zr atoms are primarily located around Al atoms.

#### 2.2.2. Structural Stability of Zr-Substituted SOD

The values of the electrostatic potential (*ν*) of Zr-substituted SOD frameworks with Si/Al ratios from 11 to one are also calculated, and the corresponding results are listed in Table 2. The *ν* of SOD-Zr_1_ is largest for the Zr-substituted SOD framework with Si/Al = 11, whereas there are no obvious differences in the values of *ν* among the other SOD-Zr_n_ (n = 2~6) structures, the values of which are similar to that of the SOD framework with Si/Al = 11 (Table 2). The calculated results indicate that the substitution of one Zr atom in the SOD framework with Si/Al = 11 enhances structural stability, whereas substitution of more Zr atoms has little effect on structural stability. For the Zr-substituted SOD frameworks with Si/Al ratios from five to one, it can be concluded that *ν* increased with an increase in the Zr content in the framework. These results demonstrate that substitution of more Zr atoms in the SOD framework enhance structural stability. As compared with *ν* of the SOD framework with Si/Al ratios from five to 1.4, low Zr content (less than two) reduces structural stability, whereas high Zr content enhances structural stability. For the SOD framework with Si/Al = 1, *ν* values of the SOD-Zr_5_ and SOD-Zr_6_ frameworks are larger than that of the SOD framework of Si/Al = 1. In conclusion, the effect of Zr substitution on structural stability is related to the Si/Al ratio; specifically, more Zr atoms in the framework enhance structural stability except for the SOD framework with Si/Al = 11.

### 2.3. Distribution of Zr and Structural Stability of Zr-Substituted AlPO-34 and SAPO-34

#### 2.3.1. Stable Location of Zr Atoms in AlPO-34 Framework

In this section, distribution of Zr atoms in the AlPO-34 (Zr^4+^ substituted for P^5+^) are investigated in the same manner as described above. There is only one crystallographically distinguishable T site in the AlPO-34 framework. Therefore, the 18 P atoms in the AlPO-34 framework are equal, and the location of the first Zr atom is random. A Brönsted acid site can be created by replacing a P atom with a Zr atom and by attaching a proton to the bridging oxygen between the proximate Zr and Al atoms. There are four different O atoms around the Zr-substitution site in the Zr-substituted AlPO-34. The H proton can be located at four different O sites, and this distribution is calculated. The results indicate that there are also no obvious relative energy differences among these different O sites for the H proton in the Zr-substituted AlPO-34 (∆E < 0.02 eV), and this is similar to the results for the SOD. Therefore, the distribution of the H proton among these different O sites is not considered in the following calculations. The stable configuration of AlPO-34-Zr_1_ is shown in Appendix A in the Appendix A; the Zr atom is at the T10 site, which is the junction of two eight-membered rings and one four-membered ring.

On the basis of the stable configuration of AlPO-34-Zr_1_, the distributions of the second Zr atom in the AlPO-34 framework are studied. There are 17 possible T sites for the location of the second Zr atom and the relative energies of these 17 possible locations are calculated. The most stable configuration of AlPO-34-Zr_2_ is shown in Appendix A, which shows that the most stable site for the second Zr atom is the T9 site (as labeled in Figure 7b). Furthermore, two Zr atoms are in the diagonal positions of the four-membered ring that are connected to the two double six-membered rings (D6R). This is consistent with the results for the most stable positions of the two Zr atoms in the Al-substituted SOD structure, as discussed in the previous section. In addition, the distribution of the third ~ sixth Zr atoms are also investigated and determined. The relative energies of the corresponding Zr-substituted AlPO-34 with Zr at different sites are plotted in Figure 3, and the most stable configurations of AlPO-34-Zr_n_ (n = 3~6) are shown in Appendix A. The most favorable sites for the third, fourth, fifth, and sixth Zr atoms are T17, T18, T3, and T16, respectively. As shown in Appendix A, the third ~ sixth Zr atoms lie in the four-membered ring that is adjacent to D6R. The results indicate that the multiple Zr atoms are distributed in a manner in which the Zr atoms are separated by the Al atom. These results are different from the distribution of Zr for which a Zr-island formed in the SOD framework. Similarly, the distributions when more than six Zr atoms are substituted are not discussed. This is also attributed to the ionic radii of the Zr atom, which is larger than that of the P atom, and this can directly affect distortion of the [PO_4_] tetrahedra with respect to the regular tetrahedra as a result of strains that are generated by the [ZrO_4_] tetrahedra in the framework.

#### 2.3.2. Stable Location of Zr Atoms in SAPO-34 Framework

The distribution of Zr atoms in the SAPO-34 framework (Zr^4+^ substituting P^5+^) is also investigated. There are 17 different possible configurations for replacing a P atom with the first Zr atom. All of the possible configurations are tested, and the most stable configuration is shown in Appendix A in the Appendix A. A plot of the relative structural energies of these 17 configurations (for which the benchmark is the energy of the structure that has the substitution at the T14 site) is shown in Figure 4. As shown in Figure 4, the preferred location for the first Zr atom is T14, and this structure has the lowest structural energy. In addition, the most stable configuration of SAPO-34-Zr_1_ is shown in Appendix A, which shows that the Zr and Si atoms lie in the diagonal positions of the same four-membered ring.

On the basis of the most stable configuration of SAPO-34-Zr_1_, the location of the second ~ sixth Zr atoms in the SAPO-34 framework are investigated. From the relative structural energies, which are plotted in Figure 4, the most suitable positions for the second, third, fourth, fifth, and sixth Zr atoms are T16, T10, T11, T8, and T17, respectively. The most stable configurations of SAPO-34-Zr_n_ (n = 2~6) are shown in Appendix A, which shows that the second Zr atom is far away from the four-membered ring where the first Zr atom is located. However, the third ~ sixth Zr atoms lie in the diagonal positions of the different four-membered ring that is adjacent to the four-membered ring in which the Si atom is located. The results indicate that the multiple Zr atoms are also distributed in the form that is dispersed rather than centralized. This is consistent with the results for the distribution of Zr atoms in the Zr-substituted AlPO-34.

For Zr-substituted AlPO-34 and SAPO-34 frameworks with different contents of Zr atoms, the lowest energy structures are chosen for further investigations. The substitution difficulty of Zr atoms is then investigated, and the corresponding P/Zr substitution energies of the most stable configurations of Zr-substituted AlPO-34 and SAPO-34 frameworks with different content of Zr are listed in Table 4; in Table 4, the benchmark is the energy of the AlPO-34 framework. The P/Zr substitution energy for the first substitution of a Zr atom in the AlPO-34 framework is 3.72 eV. The calculated result indicates that incorporating the first Zr atom into the AlPO-34 framework is unfavorable. However, as shown in Table 4, the average P/Zr substitution energy (E_av-sub_), the results indicate that substituting more Zr atoms in the AlPO-34 framework is much easier than substituting the first Zr atom.

The substitution energy for substituting a Si atom into the AlPO-34 framework is 2.80 eV. The calculated results indicate that it is easier to incorporate Si into the AlPO-34 framework than to incorporate a Zr atom. The P/Zr substitution energy of the first Zr substitution and of further Zr substitutions into the SAPO-34 framework is all reduced, and the P/Zr substitution energy of substituting two Zr atoms into the SAPO-34 framework is the smallest (0.19 eV). It can be concluded that substitution of the first heteroatom (Si or Zr) into the AlPO-34 framework is unfavorable, but the subsequent substitutions of Zr atoms become much easier. Therefore, it can be predicted that if a Zr source with other material that contains heteroatoms are co-fed during the synthesis process, incorporation of Zr into the framework should be easier than that direct synthesis of Zr-modified AlPO-34/SAPO-34.

#### 2.3.3. Structural Stability of Zr-Substituted AlPO-34 and SAPO-34

Electrostatic potentials (*ν*) of Zr-substituted AlPO-34/SAPO-34 frameworks that have different contents of Zr are also calculated, and the results are listed in Table 5. As shown in Table 5, the values of *ν* for Zr-substituted AlPO-34/SAPO-34, for which the content of Zr is less than three, are all smaller than that of the unsubstituted AlPO-34/SAPO-34 framework. However, the values of *ν* for Zr-substituted AlPO-34/SAPO-34, for which the content of Zr is more than three, are larger than that of the unsubstituted AlPO-34/SAPO-34 framework. The calculated results suggest that when the content of Zr atoms is high (more than three Zr atoms), the structural stability is enhanced. It is clear that for both the AlPO-34 and SAPO-34 frameworks, the value of *ν* increased with an increase in the content of Zr atoms in the framework, and this is consistent with the Zr-substituted SOD framework.

### 2.4. Structural Distortion of the Most Stable Zr-Substituted Frameworks

It is important to investigate the geometric perturbation induced by incorporation of the heteroatom into the SOD, AlPO-34, and SAPO-34 frameworks. The local coordination environment around the heteroatoms strongly depends on the location of isomorphous substitutions. The substitution of Si atom in SOD and that of P atom in AlPO-34 and SAPO-34 frameworks by Zr atoms lead to the expansion of corresponding tetrahedron. Therefore, it is important to investigate the influence of Zr substitution in SOD, AlPO-34, and SAPO-34 frameworks on the structural distortion. The structural distortion can be measured by two parameters, Θ and Ω, which represents the local structural perturbations of [TO_4_] and the changes in local [TO_4_] geometries caused by the heteroatom substitution. The calculation equations are given in the Appendix A in detail.

#### 2.4.1. Structural Distortion of the Most Stable Zr-Substituted SOD Framework

First, distortion of [AlO_4_] from a regular tetrahedron (Θ) and the associated Ω parameters of the SOD with Si/Al equals 11, five, three, two, 1.4, and one are investigated. The Θ and the Ω parameters for the SOD with Si/Al ratios from 11 to one are calculated, and the results are given in Table 6. As shown in Table 6, the geometric distortion of the [AlO_4_] tetrahedron is slightly affected when a high content of Al atoms is incorporated into the SOD framework. In particular, in the optimized most stable structures, the smallest distortion is obtained for the SOD with Si/Al = 3. The results reveal that there are different degrees of distortion from a regular tetrahedron (Θ) and that there are a variety of associated Ω parameters for the SOD with respect to different Si/Al ratios. These differences should be ascribed to the differences in the ionic radii of Si and Al.

On the basis of the most stable configurations of SOD with Si/Al ratios from 11 to one, Θ and the Ω parameters for the incorporation of Zr atoms into the SOD are investigated using a formula similar to that discussed above. Incorporating Zr atoms into the SOD framework results in further distortion of the [ZrO_4_] tetrahedra with respect to the regular tetrahedra. This is evidenced by the increase in the Zr-O distance and by the modification of the angles caused by Zr substitution. Incorporating Zr atoms into the SOD framework has an obvious influence on the optimized Zr-O bond lengths. This is attributed to the large ionic radius of Zr ($R_{Zr^{4+}}$ = 0.79 Å) and accordingly to the longer Zr-O bonds between the Zr atoms and the lattice oxygen atoms. The average Zr-O distances are about 2.0 Å for the SOD-Zr_n_, and this is in good agreement with the reported data for the Zr-BEA zeolites (about 1.97 Å) [1]. In this case, the six O-Zr-O angles are perturbed differently, and this is probably a consequence of the constraints that the periodic framework imposes on the [ZrO_4_] tetrahedra. Θ and the associated Ω parameters of the structures that have one ~ six Zr atoms substituted into the SOD frameworks with Si/Al ratios from 11 to one are studied because of the obvious structural distortion of [ZrO_4_] geometries (Figure 5).

First, for the SOD with Si/Al ratios from 11 to one, Θ_Al_ and the associated Ω_Al_ are generally larger for the Zr-substituted SOD as compared with the Al-substituted SOD case. Second, Θ_Zr_ and the associated Ω_Zr_ are much larger than Θ_Al_ and Ω_Al_ of the Zr-substituted SOD framework, and this indicates that incorporating Zr atoms into the SOD framework causes much more distortion in the [AlO_4_] and [ZrO_4_] geometries. In conclusion, substitutions of both Al and Zr atoms cause geometry distortions from regular tetrahedra in the SOD framework. As compared with the SOD that have Si/Al ratios from 11 to one, the doping of tetravalent ions (Zr^4+^) increases the deviations from a regular tetrahedron, and this is probably because of the larger radius of Zr atom. With respect to the influences of the Zr content on Θ and the associated Ω parameters, the results obtained for the SOD with Si/Al equals five, three, and one indicate that when the Zr content is lower, Θ and Ω are smaller. However, the smallest Θ and associated Ω are obtained for the SOD that have Si/Al equals 11, two, and 1.4 when the three Zr atoms are incorporated into the SOD framework.

#### 2.4.2. Structural Distortion of the Most Stable Zr-Substituted AlPO-34 and SAPO-34 Frameworks

Distortion of [ZrO_4_] from a regular tetrahedron (Θ) and the associated Ω parameters of the Zr-substituted AlPO-34 and SAPO-34 are investigated. The corresponding Θ and associated Ω parameters are also calculated using a formula similar to that introduced above, and the results are given in Table 7. As shown in Table 7, the smallest geometric distortion of the [ZrO_4_] tetrahedron occurs for the structure that has the first Zr atom incorporated into the AlPO-34 and SAPO-34 frameworks. However, incorporating multiple Zr atoms into the AlPO-34 and SAPO-34 frameworks results in further distortion of the [ZrO_4_] tetrahedron. The different degrees of distortion from a regular tetrahedron (Θ) and different associated Ω parameters are observed with an increase in the Zr-O distance and with different perturbations in the angles with Zr substitution. The average Zr-O distances are about 2.0 Å for Zr-substituted AlPO-34 and SAPO-34, and this is in good agreement with the reported data for Zr-BEA zeolites (about 1.97 Å) [1]. Furthermore, as compared with the O-Si-O angle, the six O-Zr-O bond angles are perturbed when Zr atoms are incorporated, and this reveals the formation of irregular tetrahedral units in the Zr-substituted AlPO-34 and SAPO-34.

For the Zr-substituted AlPO-34 and SAPO-34, the different degrees of distortion that correspond to the different structures should be ascribed to the larger ionic radius of Zr. In all cases, the O-Zr-O angles deviate significantly from the regular tetrahedral value, and deviations for T = Zr are always large. This should be ascribed to the Zr-O bonds having larger ionic character. In general, because of the larger covalent character of the Si-O bonds, there is more directionality in. In terms of the influences of Zr contents on structural distortion of the Zr-substituted AlPO-34 and SAPO-34 frameworks, the results indicate that when the Zr content is lower, the structural distortion is smaller. However, when the Zr content is high, the structural distortions of the Zr-substituted AlPO-34 and SAPO-34 frameworks are more obvious.

### 2.5. Brönsted Acidity of the Most Stable Zr-Substituted SOD, AlPO-34, and SAPO-34

It is well known that deprotonation energy is a good measure of relative acidity for the substituted models. The strength of a Brönsted acid is rigorously defined by the energy required to separate a proton from its conjugate anion to noninteracting distances. The calculation equations are given in the Appendix A in detail. DPE values do not depend on the proton acceptor, and thus they provide an acid strength scale that is independent of the reacting or adsorbing molecules involved. Zeolites that have high deprotonation energy are poor proton donors, and hence, they have weak Brönsted acidity. On the contrary, zeolites that have low deprotonation energy are better proton donors, and hence they are more acidic.

#### 2.5.1. Brönsted Acidity of the Zr-Substituted SOD with Si/Al Ratio from Eleven to One

Values of Brönsted acid strength of SOD with Si/Al ratios of 11, five, three, two, 1.4, and one are calculated, and the results are presented in Figure 6. The calculated E_DPE_ values for SOD with Si/Al ratios from 11 to one span a very broad range (12.23~11.30 eV) depending on the structural models and on the Si/Al ratio; the results are in good agreement with the data reported by Iglesia et al. (who reported E_DPE_ values about 12.40 eV) [38]. The relationship between E_DPE_ and the Si/Al ratio of the SOD is plotted (indicated by a dashed line), and according to this relationship, with a decrease in the Si/Al ratio, E_DPE_ decreases. This shows that the acid strength increases with an increase in the Al atom content. On the whole, Al substitution increases the Brönsted acid strength of the SOD, and the acid strength is related to the Si/Al ratio.

Next, on the basis of the Al distributions, the Brönsted acid strength of the Zr-substituted SOD with different Si/Al ratios are studied and compared to that of the SOD with Si/Al ratios from 11 to one. The Brönsted acid strength of the Zr-substituted SOD with different Si/Al ratios changes because of Zr substitution. The E_DPE_ values for each configuration of substituted structures are theoretically determined, and the relationship between E_DPE_ and the Zr content of Zr-substituted SOD frameworks with different Si/Al ratios are shown in Figure 6 (solid line). As compared with the Brönsted acid strength of SOD with different Si/Al ratios, for the SOD that had relatively high Si/Al ratios, all of the Zr-substituted SOD exhibited weaker acidity than the parent SOD.

The influences of the Zr content on the Brönsted acid strength of the SOD that have different Si/Al ratios are also investigated. The results obtained for the Zr-substituted SOD framework indicate that an increase in substituted Zr content increases the Brönsted acid strength of the SOD when the Si/Al ratio is 11. On the contrary, with an increase in Zr content, Zr substitution decreases the Brönsted acid strength of the SOD for the SOD when the Si/Al ratio is two. In addition, when Zr is incorporated into the SOD with Si/Al = five, three, and one, the Brönsted acid strength of the SOD first increases and then decreases with an increase in Zr content. However, the Zr substitution in the SOD framework with Si/Al = 1.4 shows the opposite phenomenon. The calculated results indicated that low Zr substitution content increases the Brönsted acid strength of the SOD when the Si/Al ratio is relatively low. However, with an increase in the Zr content, E_DPE_ increases, and this shows that Brönsted acid strength decreases as compared with the SOD with Si/Al ratios from 11 to one. From the analysis above, the Brönsted acid strength decreases because of the incorporation of Zr. For the Zr-substituted SOD, the Brönsted acid strength of the SOD with a relatively high Si/Al ratio exhibited weaker acidity than that with a low Si/Al ratio. In contrast, incorporating more Zr atoms into the SOD with different Si/Al ratios has different influences on the acid strength of zeolites. These results demonstrate that incorporating metal ions can adjust the acid strength and distribution of acid sites.

#### 2.5.2. Brönsted Acidity of the Zr-Substituted AlPO-34 and SAPO-34

In this section, the Brönsted acid strength of the Zr-substituted AlPO-34 and SAPO-34 are studied and compared. The E_DPE_ values for each configuration of substituted structures are also theoretically determined, and the relationship between E_DPE_ and the Zr content of Zr-substituted AlPO-34 and SAPO-34 frameworks are shown in Figure 6. As shown in Figure 6, both the Brönsted acid strength of the Zr-O(H)-Al acid site of the Zr-substituted AlPO-34 and SAPO-34 first decrease and then increase with an increase in Zr content. Moreover, for the Zr-substituted AlPO-34 and SAPO-34, all of the Zr-substituted SAPO-34 exhibited stronger acid strength at the Zr-O(H)-Al acid site than the Zr-substituted AlPO-34. As shown in Figure 6, the Brönsted acid strength of the Si-O(H)-Al acid site of the Zr-substituted SAPO-34 increases with the incorporation of Zr. These results are consistent with the experimental results reported by Aghaei et al. [39] who considered that incorporating Zr leads to an increase in the concentration and strength of acid sites. In other words, they found that the Zr-containing catalyst exhibits much stronger acid sites as compared with SAPO-34. In the present work, the presence of Zr in the framework increases the Brönsted acid strength of the Si-O(H)-Al acid site in the framework, and the generated acid sites (Zr-O(H)-Al acid site) and their acid properties change correspondingly.

## 3. Computational Details

### 3.1. Model

With respect to frameworks that are more interesting from a catalytic point of view, systems such as SOD and CHA are computationally very convenient because they have a relatively small unit cell and high symmetry [40]. These features permit one to fully optimize the geometry of the Zr-doped compounds and, then, to properly assess structural stability and Brönsted acidity of different concentrations of Zr. In this work, the periodic models of SOD and AlPO-34/SAPO-34 were used to investigate the preferential location sites of Zr when there are different amounts of Zr atoms and to study relative structural stability and Brönsted acidity of these substituted zeolites. The purely siliceous unit cell of the SOD is composed of 12 Si and 24 O atoms, and there is only one crystallographically distinguishable T site, as shown in Figure 7. The optimized lattice parameters (a = b = c = 8.67 Å) are in very good agreement with those from the International Zeolite Association (IZA) database (a = b = c = 8.96 Å) [41]. The CHA topological framework is initially modeled using a unit cell containing 36T (18 Al, 18 P, and 72 O atoms). The framework has a hexagonal cell with the lattice parameters a = b = 13.85 Å and c = 15.08 Å, and these are similar to parameters listed in the IZA database (a = b = 13.68 Å and c = 14.77 Å) [41]. As shown in Figure 7, there is also only one crystallographically distinguishable T site in the AlPO-34 framework. SAPO-34 is obtained by replacing phosphorus with silicon in the AlPO-34 framework. In the case of a charge imbalance (Si^4+^ substituted for P^5+^), a proton is attached to an oxygen atom that is adjacent to the isomorphous substituent site. SAPO-34 is also represented by a 36T hexagonal cell that has one Brönsted acid site (Figure 7). It should be noticed that 12 Si atoms in the SOD framework and all of the P atoms in the AlPO-34/SAPO-34 framework are labeled as Tn (where n = 1, 2, 3… 18) to distinguish different locations of Zr substitution sites.

### 3.2. Method

The Vienna ab initio simulation package (VASP) [42,43] based on plane wave basis sets was used for all the calculations with the Perdew–Burke–Ernzerhof functional (PBE) [44]. The projector augmented wave (PAW) method [45,46] was applied to describe the electron-ion interactions with cutoff energy of 500 eV for SOD system and 400 eV for AlPO-34/SAPO-34 system. The density functional dispersion correction (DFT-D) method [47] was used to take into account the dispersive interactions that influences the relative stability order of zeolite structure with heteroatom substitution. The Brillouin zone was sampled with a 3 × 3 × 3 k-point mesh for the SOD system and a 1 × 1 × 1 k-point mesh for the AlPO-34/SAPO-34 system, that generated by the Monkhorst–Pack method [48]. For the structure optimizations, all the atoms and the cell parameters of the models were fully relaxed. The convergence criterion was set to be 0.05 eV/Å for the atomic forces.

## 4. Conclusions

In conclusion, the present DFT study includes van der Waals dispersion corrections has been performed on the Zr substitution in SOD, AlPO-34, and SAPO-34. The calculated results show that the distributions of Zr in the SOD, AlPO-34, and SAPO-34 are not random. Zr atoms are more easily located at locations neighboring Al sites in the SOD framework with Si/Al ratios from 11 to one. With an increase in the number of Zr atoms, the Zr atoms primarily distribute in a form where the Zr atoms are centralized around the Al atom. However, the Zr atoms are more easily distributed in a dispersed form in the AlPO-34 and SAPO-34 frameworks. This should be ascribed to the different topological frameworks of SOD and AlPO-34/SAPO-34. The calculated electrostatic potentials (*ν*) indicate that higher Zr content generally enhances structural stability in the SOD (more than two Zr atoms) and AlPO-34/SAPO-34 (more than three Zr atoms) frameworks. In terms of the influences of Zr contents on the substitution difficulty, the results for the Si/Zr or P/Zr substitution energies of the Zr-substituted SOD, AlPO-34, and SAPO-34 frameworks indicate that SOD is the more favorable framework for Zr substitution, whereas Zr substitution in the AlPO-34 and SAPO-34 frameworks is less favorable. It should be noted that the P/Zr substitution energies for more Zr substitution in the AlPO-34 and SAPO-34 frameworks decreased as compared with substitution energy of the first Zr atom, and this means that it is possible to induce Zr atom substitution during synthesis by adding other materials that contain other heteroatoms that can be easily incorporated into the AlPO-34 and SAPO-34 frameworks.

The Brönsted acid strength of zeolite is related to the Si/Al ratio of the framework structure. For SOD with Si/Al ratios from 11 to one, when Si/Al ratio is higher, the Brönsted acidity is stronger. Except for structures that have low Zr substitution content in SOD with lower Si/Al ratios, the Brönsted acid strengths of the SOD with different Si/Al ratios decrease because of the incorporation of Zr. In this work, the acid strength of the Zr-O(H)-Al acid site in AlPO-34/SAPO-34-Zr_2_ is weaker as compared with AlPO-34/SAPO-34-Zr_1_, and then the acid strength increases with an increase in Zr content (more than two Zr atoms) in the framework. The acid strength of the Si-O(H)-Al acid site in the SAPO-34 framework increased with an increase in Zr content. Moreover, the Zr-substituted SAPO-34 exhibited stronger acid strength than the Zr-substituted AlPO-34. These results demonstrate that incorporating metal ions can change the acid strength and the distribution of acid sites. Accordingly, zeolites with different Zr distributions can be synthesized via experimental methods to obtain specific acid strength.

## Figures and Tables

**Figure 1 molecules-24-04466-f001:**
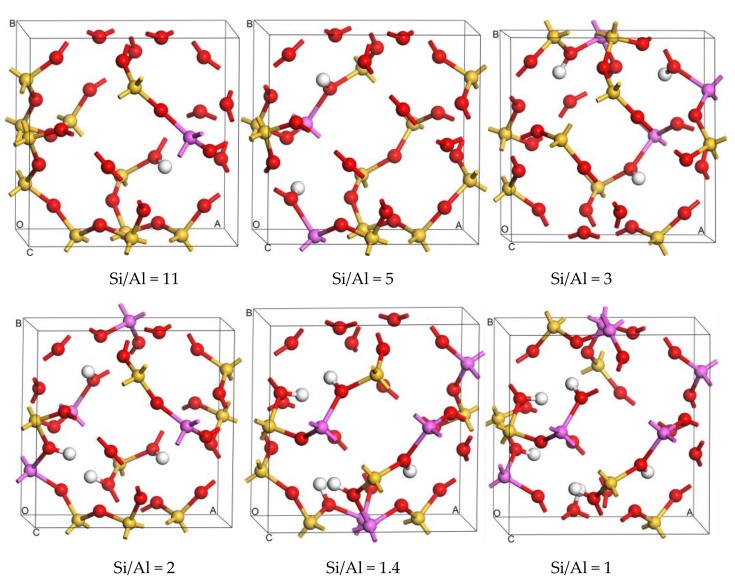
The optimized most stable configurations of SOD of different Si/Al ratio.

**Figure 2 molecules-24-04466-f002:**
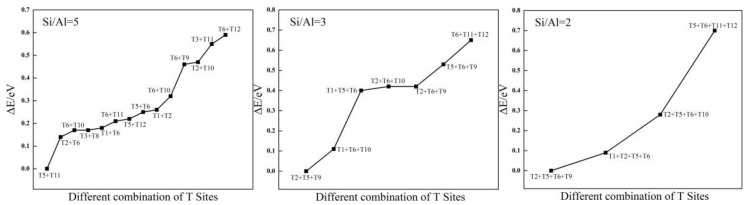
The relative energies (∆E/eV) of different structures with Si/Al ratio from 5 to 2.

**Figure 3 molecules-24-04466-f003:**
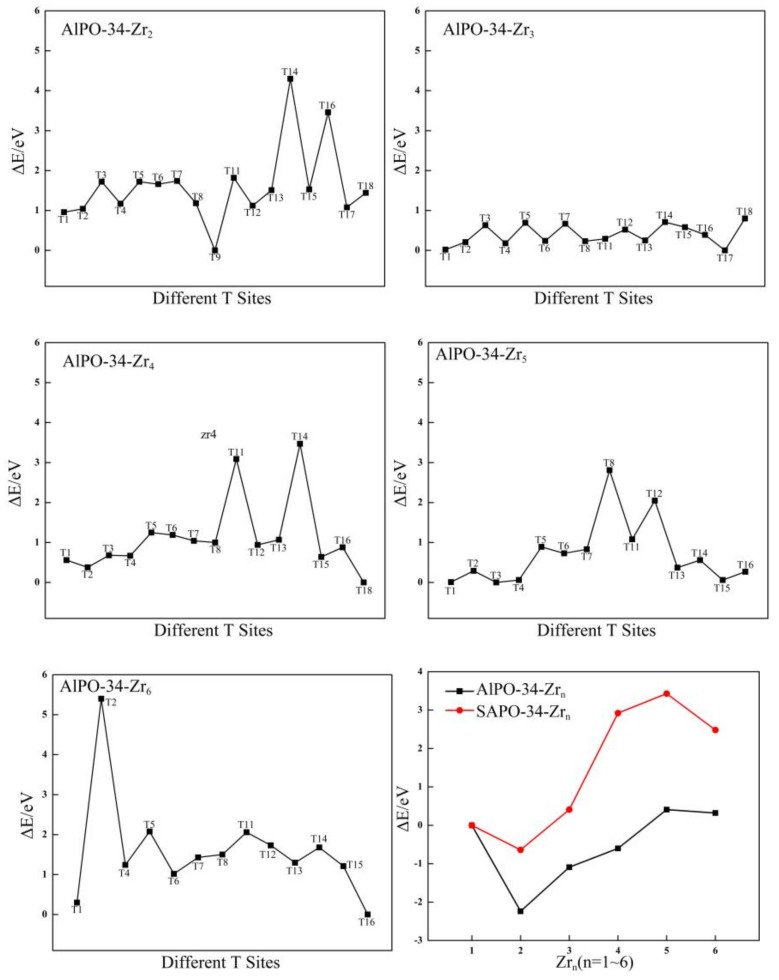
The relative energies (eV) of different T sites for Zr-substituted AlPO-34 and the relationship between the relative energies (eV) and different Zr contents for Zr-substituted AlPO-34 and SAPO-34.

**Figure 4 molecules-24-04466-f004:**
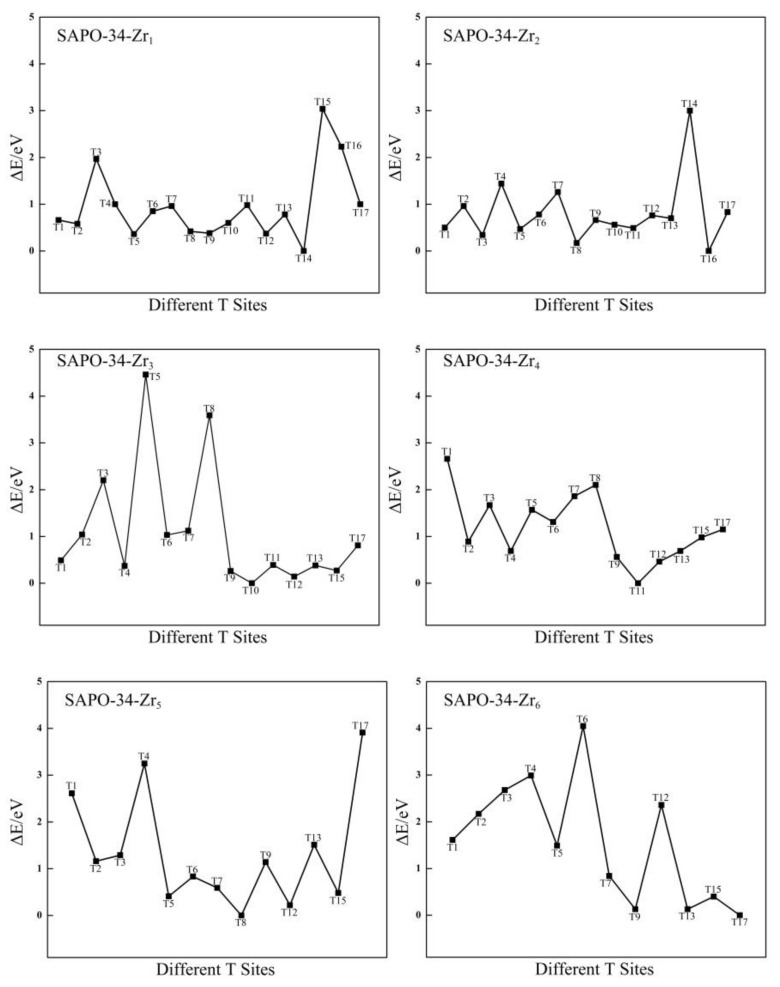
The relative energies (eV) of different T sites for Zr-substituted SAPO-34.

**Figure 5 molecules-24-04466-f005:**
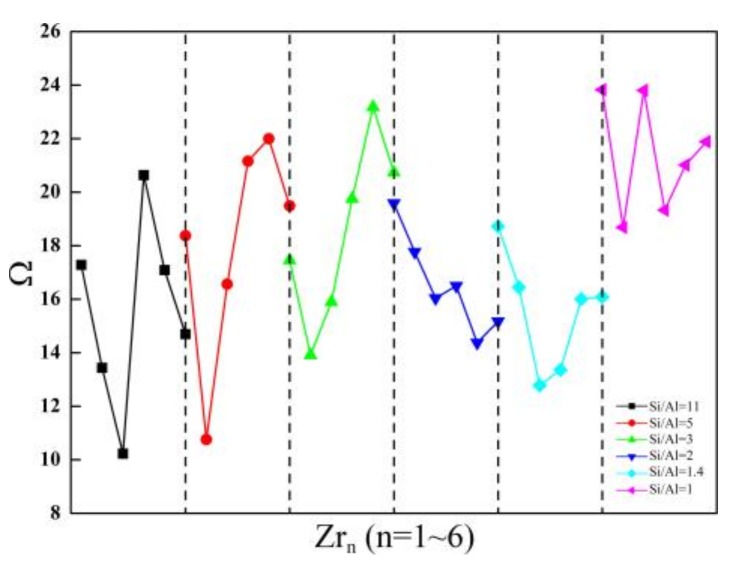
The relationship between the Ω parameters and the Zr contents of Zr-substituted SOD.

**Figure 6 molecules-24-04466-f006:**
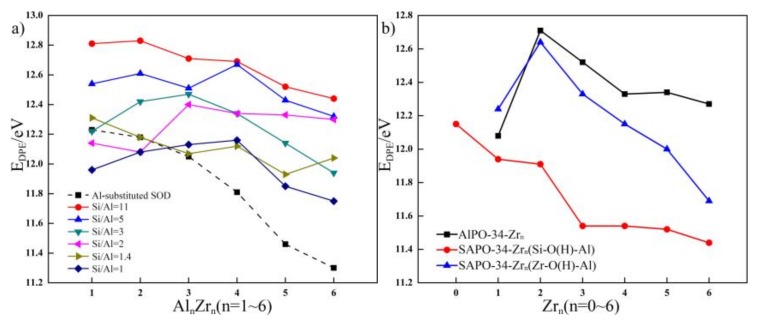
(**a**) The relationship between the E_DPE_ and Si/Al ratio of SOD, the E_DPE_ and different Zr content of Zr-substituted SOD with Si/Al ratio from 11 to 1 and (**b**) the relationship between the E_DPE_ and different Zr content of Zr-substituted AlPO-34 and SAPO-34.

**Figure 7 molecules-24-04466-f007:**
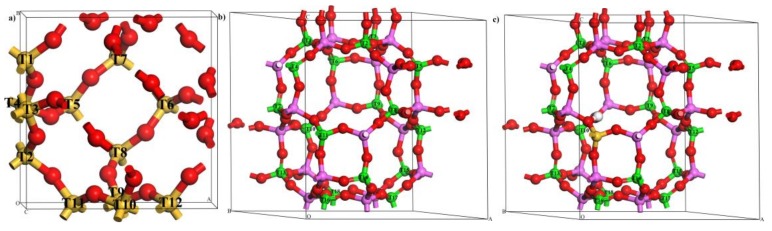
The optimized periodic model of (**a**) sodalite (SOD), (**b**) alumino-phosphate-34 (AlPO-34), and (**c**) silico-alumino-phosphate-34 (SAPO-34) in ball and stick display style. Color code: Silicon (yellow), aluminum (pink), oxygen (red), phosphorus (green), and hydrogen (white).

**Table 1 molecules-24-04466-t001:** The average substitution energy (E_av-sub_, eV) of Al atoms of SOD framework of Si/Al = 11, 5, 3, 2, 1.4 and 1.

Framework	T Sites	E_av-sub_/eV
Si/Al = 11	---	0.17
Si/Al = 5	T5 + T11	−0.63
Si/Al = 3	T2 + T5 +T9	−0.85
Si/Al = 2	T2 + T5 +T6 + T9	−0.94
Si/Al = 1.4	T1 + T2 + T6 + T9 + T10	−0.92
Si/Al = 1	---	−0.96

**Table 2 molecules-24-04466-t002:** The electrostatic potential (*ν, A*) for SOD framework with Si/Al ratios from 11 to 1 without and with Zr atoms substitution.

Zr Content	Si/Al = 11	Si/Al = 5	Si/Al = 3	Si/Al = 2	Si/Al = 1.4	Si/Al = 1
0	−9.35	−9.33	−9.39	−9.40	−9.34	−9.36
1	−9.51	−9.30	−9.27	−9.22	−9.11	−9.15
2	−9.32	−9.38	−9.35	−9.33	−9.13	−9.28
3	−9.30	−9.45	−9.39	−9.38	−9.38	−9.27
4	−9.33	−9.44	−9.46	−9.48	−9.38	−9.32
5	−9.31	−9.51	−9.52	−9.51	−9.41	−9.39
6	−9.44	−9.57	−9.63	−9.54	−9.47	−9.42

**Table 3 molecules-24-04466-t003:** The average substitution energies (eV) for Zr atoms substitution on the basis of the most stable SOD framework with Si/Al ratios from 11 to 1.

Zr Content	Si/Al = 11	Si/Al = 5	Si/Al = 3	Si/Al = 2	Si/Al = 1.4	Si/Al = 1
	E_av-sub_	E_av-sub_	E_av-sub_	E_av-sub_	E_av-sub_	E_av-sub_
1	−0.56	−0.65	−0.83	−0.83	−2.10	−2.22
2	−0.80	−0.81	−0.80	−0.87	−1.46	−1.55
3	−0.62	−0.73	−0.84	−0.90	−1.07	−1.40
4	−0.74	−0.79	−0.70	−0.87	−0.96	−0.99
5	−0.75	−0.71	−0.88	−0.75	−0.79	−0.85
6	−0.68	−0.74	−0.77	−0.62	−0.79	−0.87

**Table 4 molecules-24-04466-t004:** Average substitution energies (E_av-sub_, eV) for Zr-substituted AlPO-34/SAPO-34 frameworks.

Zr Content	AlPO-34	SAPO-34
	E_av-sub_	E_av-sub_
0	---	2.80^1^
1	3.72	0.83
2	0.74	0.10
3	0.88	0.41
4	0.78	0.94
5	0.83	0.85
6	0.67	0.55

^1^ The value is the substitution energy of Si substitution

**Table 5 molecules-24-04466-t005:** The electrostatic potential (*ν, A*) for AlPO-34/SAPO-34 framework without and with Zr atoms substitution.

Zr Content	AlPO-34	SAPO-34
0	−9.12	−9.17
1	−9.01	−8.95
2	−9.11	−9.05
3	−9.15	−9.11
4	−9.20	−9.21
5	−9.22	−9.26
6	−9.27	−9.31

**Table 6 molecules-24-04466-t006:** The distortion of [AlO_4_] from a regular tetrahedron (Θ/°) and the associated Ω parameters of SOD framework with Si/Al ratio from 11 to 1.

Framework	Θ_Al_/°	Ω_Si__→Al_
Si/Al = 11	6.18	5.79
Si/Al = 5	5.23	4.75
Si/Al = 3	4.70	4.17
Si/Al = 2	5.66	5.22
Si/Al = 1.4	6.46	6.11
Si/Al = 1	5.99	5.59

**Table 7 molecules-24-04466-t007:** The deviations of [ZrO_4_] tetrahedron caused by Zr atoms substitution in the AlPO-34 and SAPO-34.

**Framework**	**Θ_Zr_/°**	**Ω_P_** **_⟶Zr_**
AlPO-34-Zr_1_	3.06	1.51
AlPO-34-Zr_2_	12.66	9.38
AlPO-34-Zr_3_	11.28	8.25
AlPO-34-Zr_4_	13.07	9.71
AlPO-34-Zr_5_	14.43	10.83
AlPO-34-Zr_6_	13.27	9.88
**Framework**	**Θ_Zr_/°**	**Ω_P_** **_⟶Zr_**
SAPO-34	6.01(**Θ_Si_**)	3.93(**Ω_P__⟶Si_**)
SAPO-34-Zr_1_	5.85	3.80
SAPO-34-Zr_2_	13.53	10.09
SAPO-34-Zr_3_	13.17	9.80
SAPO-34-Zr_4_	12.52	9.26
SAPO-34-Zr_5_	12.18	8.98
SAPO-34-Zr_6_	13.55	10.11

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
