# Peer review of "Theoretical Study of Zirconium Isomorphous Substitution into Zeolite Frameworks"

_molecules, 2019, doi:10.3390/molecules24244466_

Round 1
Reviewer 1 Report
This paper addresses the theoretical study on effects of the substitution of different numbers of Zr atoms into zeolites. From these computations at the PBE level, the authors analyze the structural stability and structural distortion using the substitution energy and local structural perturbation parameter. Moreover deprotonation energies are used to characterize the Brönsted acidity of the most stable configurations with different amounts of Zr atoms.
The structure of this paper is not adequate because it is not clearly written and organized. Some subsections are difficult to understand and English should be improved. This current version is not sufficiently clear to deserve publication as it is.
1) A great part of this paper should be transferred to the supplementary file in order to make the text easier to read.
2) Some acronyms are not defined the first time they appear in the text (SOD, CHA, AIPO-34, SAPO-34...).
3) The word zeolite should appear in the title and in the abstract of this article. The term “molecular sieve framework “ is too general.
4) Line 12: Change GGA-PBE-D by PBE (it is well known that it is a GGA DFT approach).
5) Method: This subsection should be rewritten
6) Justify the use of PBE with dispersion for this kind of problem and include the acronym PBE in line 122
7) What’s does acronym “p-DFT-D” mean (line 123).
8) Eqs 1, 2 and 3: the rules of stoichiometry are mandatory in the scheme of chemical reactions (reactives, arrows and products). As an example, the first term should be “SOD-Si n” instead of “SOD”.
9) Eqs. 4, 5 and 6: the substitution energies are noted with a negative number. As an example, eq 4 for n =5 E(sub -4Al . However, this fact means the substitution of five Si by five Al atoms.
10) Eq. 8. In the original paper (ref. 44) zeolite is Z, here I guess that symbol for zeolite is Ze.This is not a fortunate choice as it can be confused with symbol of Zr atoms.
11) Results and discussion: There are many sections and subsections, some of them with a single paragraph. On the other hand, some figures (5, 6 and 9) and tables should be published in the Supplementary File because they add nothing relevant to the discussion.
12) Eq. 9: A units are rather unusual. If the numbers of Tables 2 and 5 are multiplied by 0.52918, the results will be in the widely used atomic units. However I recommend to employ eV as in the rest of paper.
13) Table 3 would be more readable with 7 columns directly. This Table is discussed in section 3.2.1. Clearly the influence of the optimization of more stable configurations is very important. For values of Si/Al higher differences between those average substitutions energies are smaller, ca. 2 kcal/mol. Probably, computations with another functional could give different stable configurations and consequently different substitution energies.
14) There are repetitive sentences commenting figure 8 (lines 50 – 59). In this figure, due to the great differences between Delta E for each panel, some changes are not representative. Use the same scales in all panels to prevent misleading.
15) Eqs 11 and 12: in the left hand side a new subindex should be added for referring to Si and P. Now, it is found the same name (Omega Zr).
16) Tables 6 and 7: theta y omega should have subindices.
17) Table 7: in the figure caption ZO4 should be ZrO4.
18) Conclusion: From the first sentence. the authors seem to conclude that only including dispersion it is possible to reach a non-random distribution of Zr atoms into these molecular systems. Nevertheless, in this paper there are not calculations without dispersion.
19) References: A third part of the 51 references are before year 2000. No reference after 2017 was found. Missing data to reference 8.
.
Author Response
Dear Editor,
Many thanks for your consideration to our work. We are grateful for the reviewers’ suggestions and comments concerning our manuscript (Ref: molecules-646722). The comments have improved the manuscript effectively. We have carefully revised the manuscript accordingly. Below we present a point-by-point response to their comments.
Response to Reviewer 1 Comments
Point 1: A great part of this paper should be transferred to the supplementary file in order to make the text easier to read. 

Response 1: Some texts and Figures has been transferred to the Supplementary File to make the text easier to read. The Supplementary File has been uploaded.
Lines 132-142 has been rewritten. (See Lines 122-131)
Lines 143-163 has been transferred to Supplementary File and Equations has been corrected. Some sentences have been reorganized (See Lines 229-235 and Lines 543-550).
Lines 244-248 has been transferred to Supplementary File.
Lines 468-475 has been transferred to Supplementary File.
Figures 4, 5, 6, 7 and 9 are the most stable configuration of different structures, which has been transferred to Supplementary File. New order of other Figures is also assigned.
Point 2: Some acronyms are not defined the first time they appear in the text (SOD, CHA, AIPO-34, SAPO-34...)
Response 2: The acronyms have been defined the first time they appear in the revised manuscript. (Lines 12, 13, 14, 52, 53, 109)
Point 3: The word zeolite should appear in the title and in the abstract of this article. The term “molecular sieve framework ” is too general.
Response 3: The term “molecular sieve” has been changed to “zeolite” in the revised manuscript.
Point 4: Line 12: Change GGA-PBE-D by PBE (it is well known that it is a GGA DFT approach)
Response 4: The term GGA-PBE-D has been changed to DFT-D which is the acronym of density functional theory computation including dispersion in the revised manuscript. (Line 12)
Point 5: Method: This subsection should be rewritten
Response 5: This subsection has been rewritten in the revised manuscript. (See Lines 122-131)
Point 6: Justify the use of PBE with dispersion for this kind of problem and include the acronym PBE in line 122.
Response 6: The PBE functional method with dispersion can deal with the long-range interaction and consider the weak interaction between host and guest in the molecular sieve system. And the accuracy of DFT method with dispersion correction is greatly improved. In addition, there are several literatures (J. Phys. Chem. B 2008, 112, 2573-2579; Journal of Catalysis 344 (2016) 570–577; J. Phys. Chem. C 2013, 117, 3976−3986, Molecular Catalysis 446 (2018) 106–114.) also considered the PBE functional with dispersion to perform the calculation about the incorporation of heteroatom into the zeolite framework. Therefore, the PBE function with dispersion was used in this work.
Point 7: What’s does acronym “p-DFT-D” mean (line 123)
Response 7: The acronym “p-DFT-D” should be “DFT-D”, which means the DFT method with dispersion correction. This has been corrected in the revised manuscript. (Line 126)
Point 8: Eqs 1, 2 and 3: the rules of stoichiometry are mandatory in the scheme of chemical reactions (reactives, arrows and products). As an example, the first term should be “SOD-Si n” instead of “SOD”.
Response 8: The Eqs. 1, 2, 3 have been corrected and transferred to the Supplementary File.
Point 9: Eqs. 4, 5 and 6: the substitution energies are noted with a negative number. As an example, eq 4 for n =5 E(sub -4Al . However, this fact means the substitution of five Si by five Al atoms.
Response 9: The term (1~n) in Esub(1~n)-Al means the number of Si atoms substituted by Al atoms, and the “~” is not the minus sign. The Eqs. 4, 5, 6 have been corrected and transferred to the Supplementary File.
Point 10: Eq. 8. In the original paper (ref. 44) zeolite is Z, here I guess that symbol for zeolite is Ze.This is not a fortunate choice as it can be confused with symbol of Zr atoms.
Response 10: The symbol for zeolite in Eq. 8 has been changed and transferred to the Supplementary File.
Point 11: Results and discussion: There are many sections and subsections, some of them with a single paragraph. On the other hand, some figures (5, 6 and 9) and tables should be published in the Supplementary File because they add nothing relevant to the discussion.
Response 11: The texts have been reorganized and some texts and figures have been transferred to the Supplementary File (See Response 1).
Point 12: Eq. 9: A units are rather unusual. If the numbers of Tables 2 and 5 are multiplied by 0.52918, the results will be in the widely used atomic units. However, I recommend to employ eV as in the rest of paper.
Response 12: In fact, A has no unit. Herein, A in Eq. 9 and Tables 2 and 5 means all the constants and corresponding unit of each term in Eq. 9 which make the comparison easier. Eq. 9 has been transferred to the Supplementary File.
Point 13: Table 3 would be more readable with 7 columns directly. This Table is discussed in section 3.2.1. Clearly the influence of the optimization of more stable configurations is very important. For values of Si/Al higher differences between those average substitution energies are smaller, ca. 2 kcal/mol. Probably, computations with another functional could give different stable configurations and consequently different substitution energies.
Response 13:
1) Table 3 has been changed to 7 columns.
2) Firstly, the PBE functional can describe weak interaction in the macromolecular systems, such as molecular sieve systems, more accurately. In addition, the PBE functional, which is used in this work, is chosen based on several relative references (J. Phys. Chem. C 2013, 117, 3976-3986; Catal. Lett. 2013, 148, 1246-1253; J. Catal. 2017, 352, 627-637 and so on). Therefore, the PBE function was used in this work and other functions was not considered.
Point 14: There are repetitive sentences commenting figure 8 (lines 50 – 59). In this figure, due to the great differences between Delta E for each panel, some changes are not representative. Use the same scales in all panels to prevent misleading.
Response 14:
1) The two repetitive sentences commenting Figure 8 have been corrected in the revised manuscript. (Line 373)
2) The scales in Figure 8 and Figure 10 have been redrawn with the same scale of ΔE to prevent misleading in the revised manuscript. (Lines 391-393 and Lines 420-422)
Point 15: Eqs 11 and 12: in the left hand side a new subindex should be added for referring to Si and P. Now, it is found the same name (Omega Zr).
Response 15: A new subindex has been added in Eqs. 11 and 12, referring to Si and P substitution by Zr. Eqs. 11 and 12 has been transferred to the Supplementary File.
Point 16: Tables 6 and 7: theta y omega should have subindices
Response 16: The subindices have been added to theta and omega in Tables 6 and 7 in the revised manuscript. (Table 6, Line 487 and Table 7, Line 541)
Point 17: Table 7: in the figure caption ZO4 should be ZrO4
Response 17: This mistake has been corrected in the revised manuscript. (Line 540)
Point 18: Conclusion: From the first sentence. the authors seem to conclude that only including dispersion it is possible to reach a non-random distribution of Zr atoms into these molecular systems. Nevertheless, in this paper there are not calculations without dispersion.
Response 18: This sentence has been rewritten in the revised manuscript to avoid misunderstanding. The DFT method with dispersion is used to consider the long-range interaction and weak interaction between host and guest in the molecular sieve system. Indeed, there are no calculations without dispersion in this paper. (Lines 610 and 611)
Point 19: References: A third part of the 51 references are before year 2000. No reference after 2017 was found. Missing data to reference 8.
Response 19: Some relevant references after 2017 have been added in the revised manuscript. And the reference 8 has been corrected in the revised manuscript.
Sincerely yours

Reviewer 2 Report
The manuscript from Dr. Li and co-workers reported the computational studied of isomorphous substitution of the Zrionium into molecular sieve framework. I should note that this manuscript clearly described and summarized to support the authors’ assertions. Finally, I recommend acceptance of the manuscript after minor revision based on the following comments:
Line 2: Should use the full word ‘Zirconium’ to replace the symbol Zr in the title. Line 23, grammatical mistake: should use plural for ‘acid sites are’. Line 42: not clear for ‘different group have been introduced …..’, should be ‘element/compound of different groups’ Line 48: suggested to use ‘M’ to replace ‘Me’ as ‘M’ is more common to represent metal. Lines 165 and 567: ‘et al.’ should be in italicAuthor Response
Dear Editor,
Many thanks for your consideration to our work. We are grateful for the reviewers’ suggestions and comments concerning our manuscript (Ref: molecules-646722). The comments have improved the manuscript effectively. We have carefully revised the manuscript accordingly. Below we present a point-by-point response to their comments.
Response to Reviewer 2 Comments
Point 1: Line 2: Should use the full word ‘Zirconium’ to replace the symbol Zr in the title.
Response 1: The title has been changed in the revised manuscript. (Line 2)
Point 2: Line 23, grammatical mistake: should use plural for ‘acid sites are’.
Response 2: This mistake has been corrected in the revised manuscript. (Line 25)
Point 3: Line 42: not clear for ‘different group have been introduced …..’, should be ‘element/compound of different groups’
Response 3: This sentence has been rewritten in the revised manuscript. (Line 44)
Point 4: Line 48: suggested to use ‘M’ to replace ‘Me’ as ‘M’ is more common to represent metal.
Response 4: The ‘Me’ has been changed to ‘M’ in the revised manuscript. (Line 51)
Point 5: Lines 165 and 567: ‘et al.’ should be in italic
Response 5: The ‘et al.’ has been corrected in the revised manuscript. (Lines 179, 559 and 603)
Sincerely yours

Round 2
Reviewer 1 Report
In this new version, the authors have addressed my main concerns. However, regarding point 19, reference 8 has not been changed, it is necessary to include the year and place of the congress. Moreover, I recommend reviewing the supplementary file: include the same title as the paper and number the pages in this file.
This manuscript is a resubmission of an earlier submission. The following is a list of the peer review reports and author responses from that submission.